# Orbital Modulation with P Doping Improves Acid and Alkaline Hydrogen Evolution Reaction of MoS_2_

**DOI:** 10.3390/nano12234273

**Published:** 2022-12-01

**Authors:** Fuyu Dong, Minghao Zhang, Xiaoyong Xu, Jing Pan, Liyan Zhu, Jingguo Hu

**Affiliations:** 1College of Physics Science and Technology, Yangzhou University, Yangzhou 225002, China; 2School of Physics and Electronic & Electrical Engineering, Huaiyin Normal University, Huai’an 223300, China

**Keywords:** orbital modulation, hydrogen evolution reaction, non-metal P doping, catalytic activity, MoS_2_

## Abstract

There has been great interest in developing and designing economical, stable and highly active electrocatalysts for the hydrogen evolution reaction (HER) via water splitting in an aqueous solution at different pH values. Transition-metal dichalcogenides (TMDCs), e.g., MoS_2_, are identified to be promising catalysts for the HER due to the limited active sites at their edges, while the large basal plane of MoS_2_ is inert and shows poor performance in electrocatalytic hydrogen production. We theoretically propose orbital modulation to improve the HER performance of the basal plane of MoS_2_ through non-metal P doping. The substitutional doping of P provides empty 3*p*_z_ orbitals, perpendicular to the basal plane, can enhance the hydrogen adsorption for acid HER and can promote water dissociation for alkaline HER, which creates significant active sites and enhances the electronic conductivity as well. In addition, 3P-doped MoS_2_ exhibits excellent HER catalytic activity with ideal free energy at acid media and low reaction-barrier energy in alkaline media. Thus, the doping of P could significantly boost the HER activity of MoS_2_ in such conditions. Our study suggests an effective strategy to tune HER catalytic activity of MoS_2_ through orbital engineering, which should also be feasible for other TMDC-based electrocatalysts.

## 1. Introduction

As a promising green renewable resource, hydrogen energy has been considered as one of the most ideal fuels to solve the energy crisis and environmental pollution [1,2,3]. Hydrogen evolution reaction (HER) through electrochemical water splitting has been proved to be an effective and feasible method for hydrogen production [4,5,6]. Pt-based nanomaterials are demanded as the best electrocatalyst for HER due to their near-zero hydrogen adsorption free energy (Δ*G*_H*_) and very small overpotential in acid electrolytes [7], but the high cost and scarcity motivate researchers to explore inexpensive and efficient alternatives [8]. Though electrochemical water splitting was first discovered in an acidic solution, the devices would be eroded and the produced hydrogen would be polluted by the acid vapor at high operating temperatures [9,10]. Hence, an alkaline condition is more favorable, owing to long durable and cheap electrolyzers and robust electrodes [8,11,12,13]. However, it is still far from large-scale industrial production because the HER efficiency in alkaline media is usually 2–3 orders of magnitude lower than that in acidic media [14], attributed to the sluggish water adsorption and dissociation dynamics.

Recently, two-dimensional MoS_2_, a material abundant in nature, shows a layered structure with weak van der Waals interactions between the individual S−Mo−S layers. Moreover, MoS_2_ has large surface to volume ratio and low charge carrier diffusion distance, as well as high catalytic activity for HER [15,16,17,18,19]. Due to its outstanding chemical properties, MoS_2_ is one of the most attractive alternatives to noble-metal-based electrocatalysts. Unfortunately, its catalytic activity is contributed by edged atoms, which account for a small fraction of the whole materials, while the majority of sites in the basal plane of MoS_2_ is catalytically inert [20,21,22,23]. Moreover, the intrinsically semiconducting MoS_2_ with a band gap of 1.9 eV results in poor electronic conductivity [24,25]. Hence, researchers adapt various strategies to activate the basal plane HER activity and improve the conductivity of MoS_2_, e.g., sulfur vacancies [26], doping [27,28,29,30], defects [31,32] and grain boundaries [33]. These methods can increase the number of active sites by building unsaturated atoms and alter the intrinsic conductivity via electric structure regulation. For example, a serial of transition-metal doping in MoS_2_ could improve the HER performance for more active sites and high conductivity [34]. However, metal doping has some challenges, such as the large atomic size of metal atoms would make the doping into MoS_2_ difficult and metallic dopants can also result in the growth of MoS_3_ phase, which is detrimental to the stability of MoS_2_ [29]. To ameliorate this problem, nonmetal doping [27,28,30], such as N, C, F and O. by substitution of S atom in MoS_2_ has been pursued, which is capable of realizing high HER efficiency through activating additional active sites and enhancing the conductivity. A recent study suggested that metallic Molybdenum phosphosulfide (MoP) with high conductivity exhibited good HER performance under alkaline conditions [35]. Actually, the MoP has a structure similar to MoS_2_. Inspired by this fact, we expect orbital modulation by doping nonmetal P atoms into MoS_2_ might be an efficient way to enhance the HER performance of MoS_2_ under acid and alkaline conditions. In fact, phosphorus, as a dopant doped into MoS_2_, has been proved to be feasible and reliable in experiments. For example, Li et al. showed that P-doped MoS_2_ nanosheets with enlarged interlayer spacing exhibited remarkable electrocatalytic activity and good long-term operational stability [36]. Bian et al. found that P-doped MoS_2_ nanosheets on carbon cloths displayed excellent catalytic activity with a low overpotential of 133 mV to drive the current density of 20 mA cm^−2^, along with good stability in acid media [37]. Huang reported that interface construction of P-substituted MoS_2_ was considered as an efficient and robust electrocatalyst for alkaline HER [38]. Our DFT results show that the orbital modulation with P-doping not only increases electronic conductivity of MoS_2_ but also activates the inert surface for more emerging active sites; as a result, the reaction barriers are effectively decreased and the HER performance can be greatly improved under both acid and alkaline conditions. The novel concept of orbital modulation can be utilized to design other HER catalysts.

## 2. Computational Model and Methods

All calculations in this study were performed by the Vienna ab initio simulation package (VASP) [39,40] with spin-polarized density function theory (DFT). The exchange correlation interaction was modeled by generalized gradient approximation with Perdew–Burke–Ernzerhof (PBE) [41] and considered van der Waals corrections. The electron–ion interactions were described by the projector-augmented plane wave pseudopotential [39]. The plane wave basis set was used to expand the wave functions with a cutoff energy of 500 eV. The Brillouin zone was sampled by the Monkhorst-Pack k-point meshes of 5 × 5 × 1 and 7 × 7 × 1 for geometric optimization and electronic structures, respectively. The convergence thresholds were set to 1.0 × 10^−5^ eV/atom for energy and 0.01 eV/Å for interatomic force. To investigate the transition states and the minimum energy pathways (MEPs) for HER, the climbing image nudged elastic band method was used, in which the convergence criterion was 0.02 eV/Å for MEP [42].

During the calculations, the vacuum thickness of monolayer MoS_2_ was set to be at least 15 Å to avoid the interaction between adjacent images. In order to study the properties of P-doped MoS_2_, we constructed a 4 × 4 × 1 supercell of MoS_2_ (see Figure 1a) containing 16 Mo atoms and 32 S atoms. We considered the structural symmetry, one, two and three P atoms, respectively, doped in MoS_2_ by substituting intrinsic S atoms, which corresponds to a doping concentration of 2%, 4% and 6%, respectively, which may be achieved in the experiments. Ye et al. reported a one-step approach to improve the HER activity of MoS_2_ and MoP via formation of an MoS_2(1−*x*)_P*_x_* (*x* = 0 to 1) solid solution [43]. Liu et al. successfully doped P atom into MoS_2_ and the concentration of P dopant was up to 5.1% [36]. To verify the reliability of PBE + vdW method, we compared the lattice constants, the bond length and band gap of monolayer MoS_2_ calculated by our PBE + vdW method with other methods and the experimental results in Ref [25]. The comparative results show that the PBE + vdW calculated results are in good agreement with the experimental values [44,45]. For example, the calculated lattice parameters are *a* = *b* = 3.17 Å, the bond length of Mo-S is *d*_S-Mo_ = 2.43 Å and the band gap is *E*_g_ = 1.74 eV, well consisting with the experimental values (*a* = *b* = 3.16 Å, *d*_S-Mo_ = 2.42 Å, *E*_g_ = 1.90 eV) and other theoretical reports [30,46].

For acid HER, two steps are involved [47] as shown in Figure 1b:H^+^ + e^−^ → H* (the Volmer reaction),
H* + H^+^ + e^−^ → H_2_ (the Heyrovsky reaction).

Adsorption free energy of hydrogen (Δ*G*_H*_) is a key indicator for evaluating HER activity, which can be defined as Δ*G*_H*_ = Δ*E*_H*_ + Δ*E*_ZPE_ − TΔ*S*_H*_, where Δ*E*_H*_, Δ*E*_ZPE_ and Δ*S*_H*_ are the hydrogen chemisorption energy, zero-point energy change and the entropy of H* adsorption. The closer the Δ*G*_H*_ is to zero, the higher HER activity is, because atomic hydrogen is in a thermoneutral state, proton/electron transfers and hydrogen release become more efficient.

For alkaline HER, the reaction process is more complex involving five steps [48,49] as shown in Figure 1c,
H_2_O + * → H_2_O*(1)
H_2_O* → H* + OH*(2)
H* + OH* → H* + OH^−^(3)
H* + H_2_O → 2H* + OH*(4)
2H* + OH* → H_2_ + OH^−^(5)
where * represents adsorbed site of the surface. The free energy can be defined as Δ*G* = *E*_ads_ + ΔZPE − *T*Δ*S*, where *E*_ads_, ΔZPE and Δ*S,* respectively, represent adsorption energy of adsorbate, the difference in zero-point energy of hydrogen vibration between the adsorbed state and the gas phase and the entropy difference between the adsorbed state of the system and gas phase at the standard condition.

## 3. Results and Discussion

First, we investigated the stability of P-doped MoS_2_ by calculating the formation energy as *E*_form_ = *E*_tot_ − *E*_DS–MoS2_ − n*μ*_S_ [45,50], where *E*_tot_ and *E*_DS–MoS2_ are energies of P-doped MoS_2_ and S-deficient MoS_2_, n and *μ*_S_ are the numbers of dopant P atoms and the chemical potential of S calculated by the binding energy of S_2_ molecule. The calculated results are −0.411, −0.906 and −2.151 eV for 1P-, 2P- and 3P-MoS_2_, respectively. The negative values indicate the doping of P is energetically favorable. Moreover, the formation energies decrease with the increasing of P dopant concentration, suggesting that P doping can be easily achieved in MoS_2_.

### 3.1. Electric Structure and Stability of P-doped MoS_2_

Figure 2a shows the electronic structure of pristine and P-doped MoS_2_. The pristine MoS_2_ is a semiconductor whose band gap is calculated to be 1.74 eV. Both the VBM and CMB are mainly composed of Mo-4*d* and S-3*p* orbitals. After one, two and three P atoms are substitutionally doped into a 4 × 4 supercell of MoS_2_ (denoted as 1P, 2P and 3P hereafter), owing to strong hybridization near the Fermi level (*E*_f_) among the Mo-4*d*, S-3*p* and P-3*p* orbitals, the band gap changes to 1.65, 1.63 and 1.76 eV, respectively. The reduced band gaps in 1P and 2P doping are conductive to thermal excitation, which can promote electric conductivity. The heavy doping, i.e., 3P doping, even causes the emergence of mid-gap states, which are constituted by Mo-4*d*, P-3*p* orbitals. The impurity states could improve the electric conductivity of MoS_2_ since the localized states can induce the inner-electric field and enhance carrier mobility. Furthermore, the orbital charge density of MoS_2_ in Figure 2b displays that the charges are uniformly distributed on Mo and S atoms, indicating that the activity of any S atom on the surface is almost the same. P doping, however, would make the surface charge redistributed and the charge densities accumulate around the dopant P (highlighted in dashed circles), in which P-3*p*_z_ orbitals, perpendicular to the basal plane, are found to be isolated and unoccupied (see blue dashed circles). These unoccupied P-3*p*_z_ orbitals can provide active sites for HER, as discussed in the following sections.

### 3.2. Acid HER on MoS_2_ Surface

To investigate HER performance, we focus on the catalytic activity of active sites. Δ*G*_H*_ is an effective descriptor for identifying catalytic activity in acid HER. For pure MoS_2_, the Δ*G*_H*_ on the surface of the S atom is as high as 1.90 eV (see Figure 3a), indicating that the basal plane is inert for HER, ascribed to energetical-unfavorable adsorption of hydrogen onto the MoS_2_ surface. As we know, the HER activity of MoS_2_ appears at its edge [20,21,22,23]. We further built the MoS_2_ nanoribbon with zigzag edges, as shown in Figure 3b, where one edge was terminated with Mo atoms and another edge was terminated with S atoms. The Δ*G*_H*_ at Mo edge is −0.28 eV and Mo acts as the active site, while the Δ*G*_H*_ at S edge is −0.38 eV and S acts as the active site, which is much lower than that on the MoS_2_ surface. Thus, catalytic activity at the edge is much higher than that on the surface.

For P doping, the Δ*G*_H*_ is −0.88 and −0.82 eV for 1P and 2P when dopant P serves as the active site. Δ*G*_H*_ < 0 means that P dopant greatly enhances the binding strength between the active site and hydrogen. Contrarily, the dissociation of hydrogen will become difficult. For the case of 3P doping, the P dopant pulls out from the surface, the catalytic activity further enhances, the Δ*G*_H*_ decreases to −0.02 eV and it is ideal for the HER. Furthermore, we doped MoS_2_ with 4P dopant for 8% concentration, the Δ*G*_H*_ is −0.19 eV with P acting as the active site. However, the Δ*G*_H*_ of intrinsic MoP is −0.54 eV [51], lower than that of 3P- and 4P-doped MoS_2_, in which the 3P doping corresponds to the optimal dopant concentration for the HER. Significantly, when S atoms, neighboring dopant P atoms, act as active sites, the Δ*G*_H*_ becomes 0.49, 0.42 and 0.59 for 1P, 2P and 3P doping. The lowered Δ*G*_H*_ as compared to the pristine MoS_2_ surface is due to the charge redistribution induced by the P dopant. Specifically speaking, more charges concentrate on neighboring S atoms, making the strength between S and hydrogen enhanced. Hence, the surface activity is stimulated and the active sites increase.

To investigate the effect of dopant P on the HER process, we further calculated bond strength between the active site and hydrogen atom by calculating Crystal Orbital Hamilton Population (COHP) and its integrated values (ICOHPs) [52,53,54,55]. The positive and negative values in the –COHP curves correspond to the contributions of orbital interaction from bonding and antibonding states. As shown in Figure 3, the ICOHP values are −3.81, −4.50, −4.50 and −4.17 for pristine, 1P-, 2P- and 3P-doped MoS_2_, respectively. A more negative ICOHP value indicates stronger bonding strength. Obviously, P doping strengthens the interaction between the active site and H atom. However, too strong bonding strength will also hamper H dissociation in the HER. Contrarily, rather weak binding makes the H adsorption unfavorable. In detail, for pristine MoS_2_, the S atoms are considered as the active site and the ICOHP value of –3.81 is large; due to the weak binding, H adsorption is difficult. Thus, Δ*G*_H*_ is a huge positive value of 1.90 eV. For 1P and 2P doping, P atoms act as the active sites and the ICOHP values are −4.50. The more negative ICOHP is responsible for the stronger bonding strength between the active site and H atoms. Hence, the Δ*G*_H*_ becomes negative, i.e., −0.88 and −0.82 eV, respectively, but hydrogen desorption is difficult. As we know, the moderate value of the ICOHP can benefit both the proton/electron transfer and hydrogen-release processes. Our calculations suggest that the ICOHP of −4.168 in 3P-MoS_2_ should be the optimal value, which leads to an almost ideal value of Δ*G*_H*_ i.e., −0.02 eV. In detail, the −COHP curves in Figure 3 show the interaction between the active site and the H atom contributes from the hybridization between H-1*s* orbital and S-3*p*_x_, S-3*p*_y_ and S-3*p*_z_ orbitals in pristine MoS_2_ surface. For 1P- and 2P-MoS_2_, the −COHP curve comes from interaction between H-1s and P-3*p*_z_ states. Apparently, the large empty *p*_z_ orbitals play an important role in hydrogen adsorption. For 3P doping, the −COHP originates mainly from P-3*p*_z_ states, with a few from P-3*p*_y_ orbitals. The *p*_y_ orbitals may contribute to desorption.

For the acid HER dynamic process, the first step of the Volmer reaction is fast. Usually, the second step coexists in both the Heyrovsky and the Tafel reactions, which is the rate-determining step. We find that S atoms are favored as active sites for the Tafel reaction and P atoms are suitable as active sites for the Heyrovsky reaction. Figure 4 shows the MEP of the Heyrovsky reaction, where a proton in the water layer reacts with adsorbed hydrogen to form H_2_. In the transition state, an adsorbed hydrogen atom gradually approaches the hydrogen atom of a hydronium ion in the water layer, the distance between the two interacting H atoms is about 3 Å and then it breaks away from the MoS_2_ surface; finally, the H_2_ molecule forms with a H-H bond length of 0.75 Å. The activation barrier is 0.65 for pristine MoS_2_, which decreases to 0.52, 0.50 and 0.43 eV in 1P-, 2P- and 3P-doped MoS_2_, respectively. The lower barriers indicate that P doping is able to accelerate HER kinetics occurring on the MoS_2_ surface.

### 3.3. Alkaline HER on MoS_2_ Surface

Figure 5 shows that the alkaline HER catalytic pathway includes double-water adsorption and dissociation on the MoS_2_ surface. The reaction is more sluggish than the acid HER [14]. Initially, H_2_O adsorbs on the top of the surface. Secondly, H_2_O dissociates to form H* and OH* (the Volmer step). Thirdly, OH^−^ moves away from the MoS_2_ surface. Fourthly, another H_2_O molecule adsorbs on top of MoS_2_. Next, two H* and a single OH* adsorbed successively on different neighboring sites. Finally, H-H separates from the surface to form H_2_ and OH^−^ (Tafel step). During the alkaline HER process, water adsorption and disassociation determine HER kinetics performance, in which active sites play an important role. Table 1 shows the active site at each step on the MoS_2_ (001) facet.

For pristine MoS_2_, the active site at each step is the S atom. As shown in Figure 5a, there is large H_2_O adsorption free energy (Δ*G*_H2O*_) of 2.39 eV and a higher H_2_O dissociation energy barrier of 4.42 eV for TS1 during the first water adsorption and dissociation. During the second water adsorption and dissociation, Δ*G*_H2O*_ is 2.31 eV, almost unchanged, and the dissociation energy barrier is still large with 3.61 eV for TS2. Thus, the basal plane of MoS_2_ remains inert in an alkaline condition.

For P doping, during the first water adsorption, P acts as the adsorbed site and H_2_O adsorption remains difficult with Δ*G*_H2O*_ of 2.32, 2.43 and 2.46 eV for 1P, 2P and 3P doping. During the first dissociation process, for 1P doping (see adsorption site in Table 1), with assistance of the neighboring S by drawing H^+^ and leaving OH^−^ on P atom, the catalytic kinetic is greatly enhanced since the dissociation energy barrier decreases greatly to 2.08 eV. Notably, for 2P and 3P doping, dissociated H^+^ adsorbs on a nearby P atom. The energy barriers dramatically drop to 0.98 and 0.13 eV. Obviously, P doping makes the water dissociation barrier greatly decrease. Thus, it will enhance alkaline HER performance because the P dopant is more attractive for OH^−^ and H^+^ than surface S atoms, which is critical to boost dissociated behaviors. In this case, the reason why the dissociation energy barrier in 3P doping is much lower than 2P doping, due to the fact that the dopant P atom is pulled out of the surface, thus, providing more active sites for HER in 3P-MoS_2_.

To figure out the effect of different active sites on H_2_O dissociation, we further calculate −COHP curves between H* and the active site in Figure 6a. The ICOHP values are −3.68 and −3.82 with S atoms acting as active sites for pristine MoS_2_ and 1P-MoS_2_ and the ICOHP comes from hybridization among H-1s and S-3*p*_x_, *p*_y_, *p*_z_ orbitals. For 2P and 3P doping, P acts as the adsorption site. The ICOHP values are −4.45 and −4.50, which come from hybridization between H-1s and P-3*p*_z_. H* prefers to stick to the P atom. P-3*p*_z_ orbitals play an important role for enhancing the interaction between H* and the active site. For the interaction between HO* and the active site, as shown in Figure 6b, the ICOHP value is −2.12 in the case of pristine MoS_2_ and the active site is S, which is much larger than the case of regarding P as adsorption site. In the latter case, the ICOHP values are −3.35, −3.09 and −3.02 for 1P, 2P and 3P doping, respectively. Obviously, HO* is difficult to adsorb on the S site. Compared to the strength between H* and the adsorption site, the interaction between HO* and the active site is weak and orbital analysis in Figure 6b shows the COHP curves come not only from P-3*p*_z_ orbitals but also from P-3*p*_y_ and P-3*p*_x_ orbitals, which is in favor of the process of step R3 to R4; that is, HO* breaks from the surface. The free energies decrease to 1.90, 0.49, −0.82 and 0.02 eV for pristine MoS_2_, 1P-, 2P- and 3P-MoS_2_, respectively.

For the second adsorption and dissociation processes, the H_2_O adsorption remains difficult, of which the free energies are as large as 2.31, 2.12, 2.28 and 1.77 eV, for pristine MoS_2_, 1P, 2P and 3P-MoS_2_, respectively. For the second H_2_O dissociation, the energy barriers are, respectively, 3.61, 2.12, 2.05 and 0.31 eV for pure, 1P-, 2P- and 3P-doped MoS_2_. For 1P doping, whether the first or second H_2_O adsorption, the active site for H* adsorption and HO* adsorption is always S and P. Thus, the barrier energy does not change much. Significantly, for 2P doping, the dissociation energy barrier at second water dissociation obviously increases compared to the first water dissociation because the active site for H* adsorption and HO* adsorption is, respectively, on different doped P atoms at the first H_2_O dissociation, while the active site for H* adsorption and HO* adsorption is, respectively, on neighboring S and dopant P atoms at the second H_2_O dissociation. Obviously, P has more affinity for H^+^ and OH^−^ than that of S. The −COHP curve in Figure 6c clearly shows the interaction between the active site and H*. For S acting as the active site, −COHP curve originates from not only S-3*p*_z_ orbitals but also from S-3*p*_x_ and S-3*p*_y_ orbitals. Thus, the strength between S and H* is weak. While P acts as the active site, the −COHP curve mainly contributes from P-3*p*_z_, displaying strong interaction between P and H*. For the interaction between HO* (see Figure 6d), the adsorption sites are all P atoms for P doping and the −COHP curve displays a similar feature.

## 4. Conclusions

In summary, based on DFT electronic-structure calculations and kinetics reaction pathway analysis, we demonstrated that orbital modulation by via P doping could successfully enhance HER performance of an MoS_2_ (001) facet in both acid and alkaline conditions. Enhancing the efficiency of hydrogen/water adsorption and dissociation plays an important role in improving HER catalytic performance. The −COHP analysis shows that the P dopant provides many empty 3*p*_z_ orbitals. These 3*p*_z_ orbitals, perpendicular to the basal plane, can enhance the hydrogen adsorption for acid HER and promote water dissociation for alkaline HER. Thus, it improves the HER catalytic kinetics effectively. As a result, 3P-doped MoS_2_ exhibits excellent HER activity in both acid and alkaline media, since the 3P doping could increase conductivity, provide more active sites and improve HER catalytic kinetics with desirable Δ*G*_H*_ and a low reaction barriers. Our work reveals the underlying mechanism of the improved catalytic activity in MoS_2_ from an atomic orbital point of view, which can be extended to other 2D TMDCs to design high-performance catalysts.

## Figures and Tables

**Figure 1 nanomaterials-12-04273-f001:**
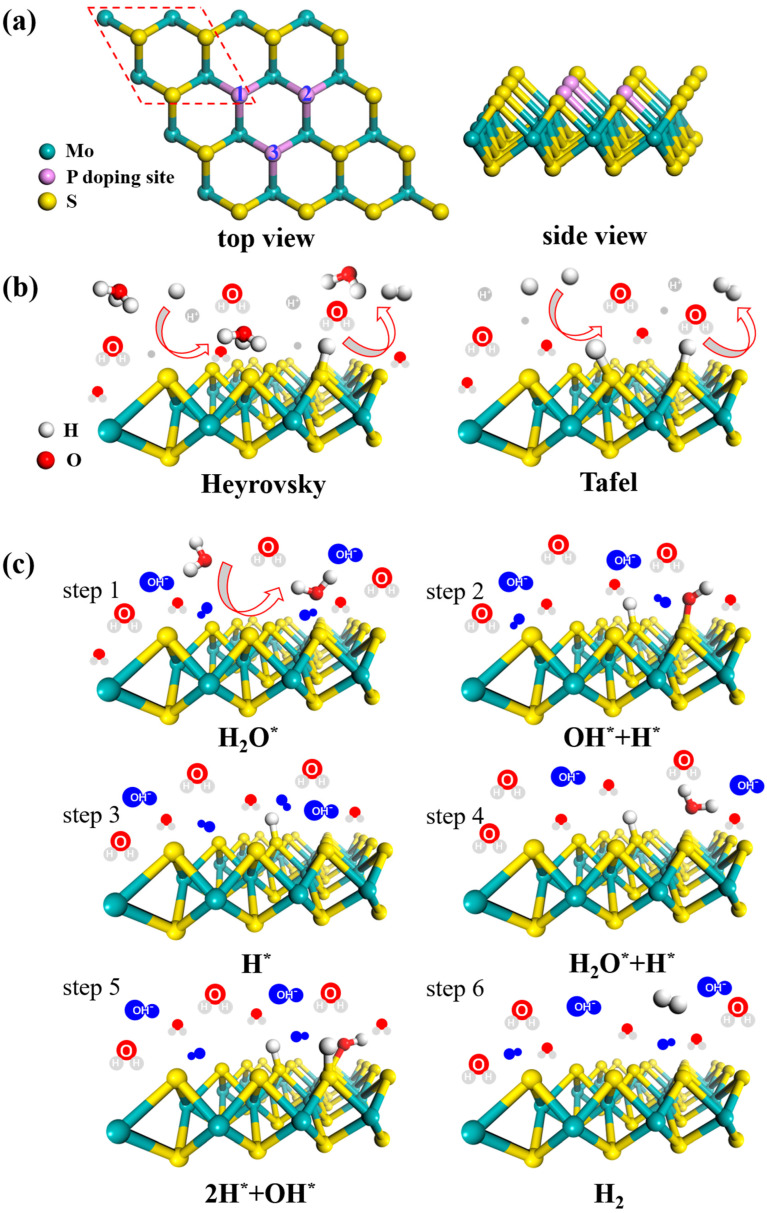
(**a**) Top and side views of MoS_2_ (001) facet and the model diagrams of HER (**b**) in acid and (**c**) alkaline conditions.

**Figure 2 nanomaterials-12-04273-f002:**
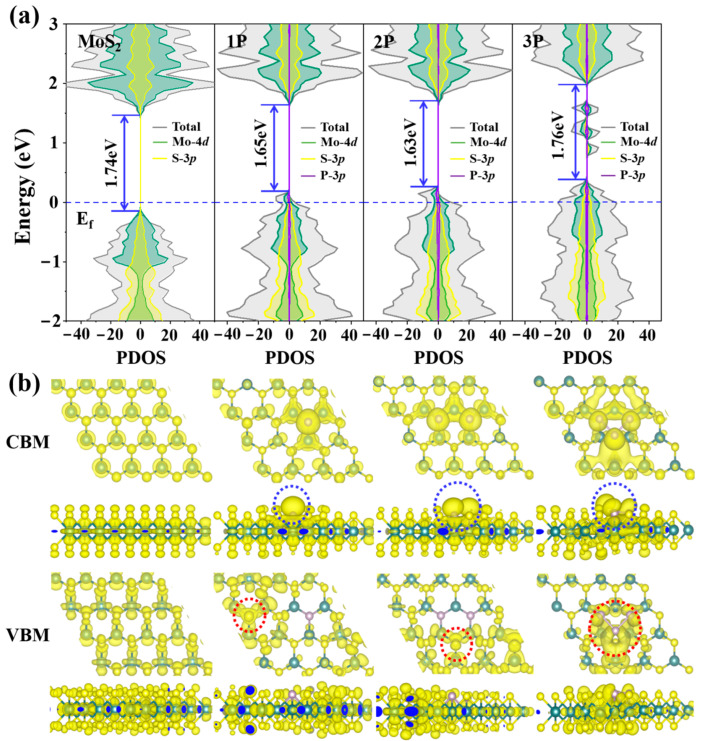
(**a**) Total and partial density of states and (**b**) charge density at CBM and VBM in pure, 1P-, 2P- and 3P-doped MoS_2_.

**Figure 3 nanomaterials-12-04273-f003:**
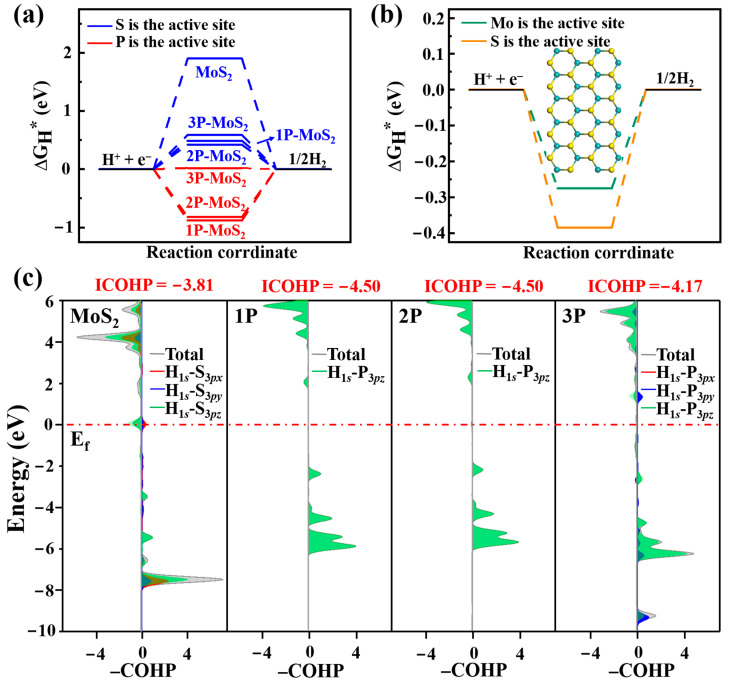
The Δ*G*_H*_ of (**a**) pure, 1P-, 2P- and 3P-doped MoS_2_ with S and P acting as active sites, the Δ*G*_H*_ of (**b**) zigzag MoS_2_ nanoribbion with Mo and S acting as active sites and (**c**) the COHP between H and active sites on pure, 1P-, 2P- and 3P-MoS_2_ in acid condition.

**Figure 4 nanomaterials-12-04273-f004:**
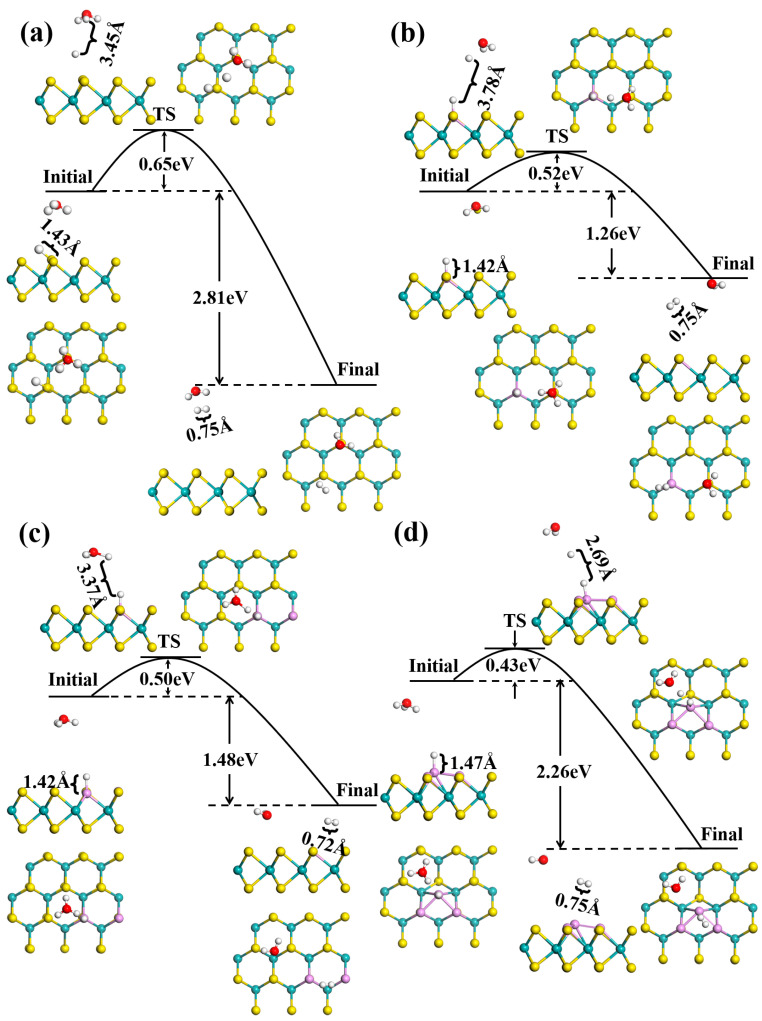
Minimum-energy path of the Heyrovsky reaction on (**a**) pristine, (**b**) 1P-, (**c**) 2P- and (**d**) 3P-doped MoS_2_.

**Figure 5 nanomaterials-12-04273-f005:**
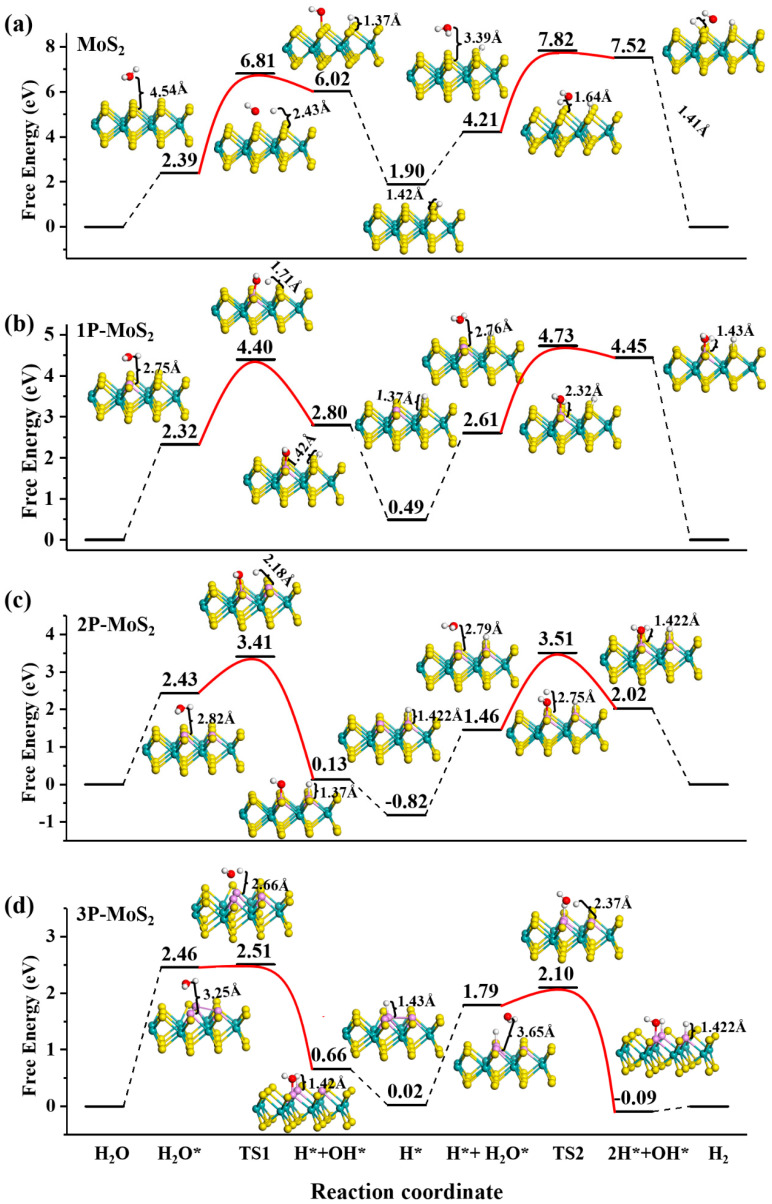
Free-energy diagram for alkaline HER on (**a**) pure, (**b**) 1P-, (**c**) 2P- and (**d**) 3P-MoS_2_.

**Figure 6 nanomaterials-12-04273-f006:**
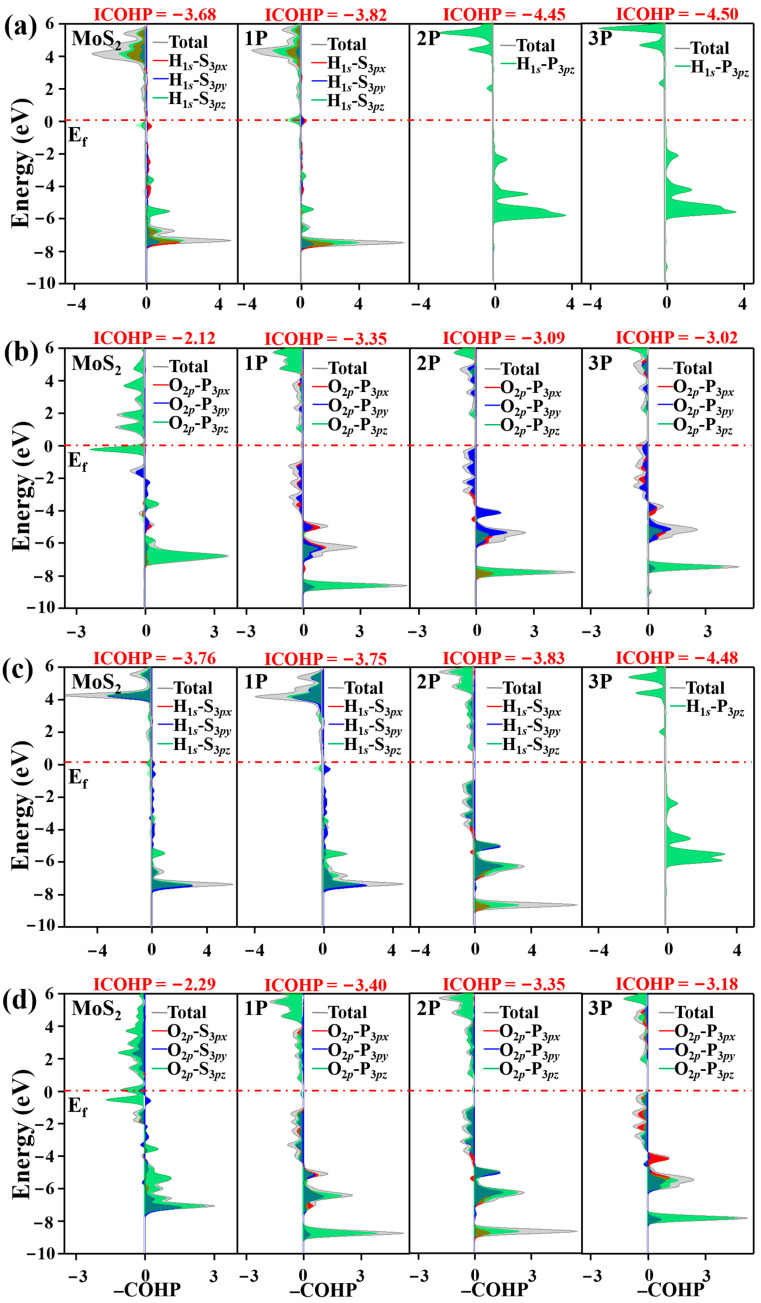
The –COHP between (**a**) H* and active sites and (**b**) HO* and active sites at the first adsorption and dissociation, (**c**) H and active sites and (**d**) HO* and active sites at the second dissociation on pristine, 1P-, 2P- and 3P-MoS_2_ in alkaline condition.

**Table 1 nanomaterials-12-04273-t001:** The steps of alkaline HER process, the active site for each step, the free energy of intermediates (Δ*G* in eV) and barrier energy of H_2_O dissociation (Δ*E* in eV) for pure, 1P-, 2P- and 3P-doped MoS_2_. * represents adsorbed site of the surface.

System	The First Step		The Second Step			The Third Step		The Fourth Step			The Fifth Step			
	H_2_O*	Δ*G*	H*	OH*	Δ*E*	H*	Δ*G*	H*	H_2_O*	Δ*G*	H_1_*	H_2_*	HO*	Δ*E*
pure	S	2.39	S	S	4.42	S	1.90	S	S	2.31	S	S	S	3.61
1P	P	2.32	S	P	2.08	S	0.49	S	P	2.12	S	S	P	2.12
2P	P	2.43	P	P	0.98	P	–0.82	P	P	2.28	P	S	P	2.05
3P	P	2.46	P	P	0.13	P	0.02	P	P	1.77	P	P	P	0.31

## Data Availability

Not applicable.

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
