# Peer review of "Orbital Modulation with P Doping Improves Acid and Alkaline Hydrogen Evolution Reaction of MoS2"

_nanomaterials, 2022, doi:10.3390/nano12234273_

Round 1

Reviewer 1 Report

Nanomaterials 204462

Dong et al. present an exciting approach to using the DFT method for catalyst optimization. The article itself is fascinating, and it is within the journal's scope.

This reviewer suggests a major English language revision of this version prior be accepted.

Also, in the attached documents, several questions should be appended before the acceptance.

Author Response

Dong et al. present an exciting approach to using the DFT method for catalyst optimization. The article itself is fascinating, and it is within the journal's scope.

This reviewer suggests a major English language revision of this version prior be accepted.

Also, in the attached documents, several questions should be appended before the acceptance.

Thank the reviewer for the positive comments. We have checked the manuscript and corrected the spelling and grammar carefully, and hope these modifications and improvements can make our manuscript more suitable for publication.

Point 1. Why would the produced hydrogen be polluted?

Reply: Thank you for the comment. In acidic solution, the produced hydrogen would be polluted by the acid vapor at high operating temperatures, which has been explained in the revised manuscript.

“the devices would be eroded and the produced hydrogen would be polluted by the acid vapor at high operating temperatures”

Point 2. Why was this approach selected? Need to be explained better.

Reply: Thank you for the suggestion. To verify the reliability of PBE+vdW method, we compare the lattice constants, the bond length and band gap of monolayer MoS2 calculated by our PBE+vdW method with HSE06 method and the experimental results. The comparative results show that the PBE+vdW calculated results are in good agreement with the experimental values. For example, the calculated lattice parameters are a = b = 3.17 Å, the bond length of Mo-S is dS-Mo = 2.43 Å, the band gap is Eg = 1.74 eV, well consisting with the experimental values (a = b = 3.16 Å, dS-Mo = 2.42 Å, Eg = 1.90 eV). Though the HSE06 functional can also give better structural parameters and band gaps for the MoS2 systems, it needs the significantly demanding computational cost. Additionally, our calculated results well agree with other theoretical reports [J. Am. Chem. Soc., 2013, 135, 17881 and Applied Surface Science, 2021, 542, 148535], thus, we adopt the PBE+vdW method in our calculations. We have added related discussions in the revised manuscript.

Table.The structural parameters of the lattice constants a, b and c (in Å) of bulk MoS2, the average bond length (in Å) of Mo–S and the band gaps of bulk MoS2 and monolayer MoS2 (in eV) calculated using the PBE+vdW and HSE06 functionals. For comparison purposes, the corresponding experimental values are also included. [Phys.Chem.Chem.Phys., 2017, 19, 24594]

PBE+vdW

HSE06

Exp.

a = b (Å)

3.17

3.16

3.16

c (Å)

12.30

12.29

12.29

dS-Mo (Å)

2.43

2.42

2.42

Eg(bulk) (eV)

1.27

1.48

1.29

Eg(ML) (eV)

1.74

2.08

1.90

“To verify the reliability of PBE+vdW method, we compare the lattice constants, the bond length and band gap of monolayer MoS2 calculated by our PBE+vdW method with other methods and the experimental results. The comparative results show that the PBE+vdW calculated results are in good agreement with the experimental values. For example, the calculated lattice parameters are a = b = 3.17 Å, the bond length of Mo-S is dS-Mo = 2.43 Å, the band gap is Eg = 1.74 eV, well consisting with the experimental values (a = b = 3.16 Å, dS-Mo = 2.42 Å, Eg = 1.90 eV) and other theoretical reports.”

Point 3. If the assumption is that P is located in such definitive locations, authors should provide some kind of justification.

Reply: Thank the review for the suggestion. To investigate the stability of P-doped MoS2, we calculate the formation energy as Eform = Etot – EDSMoS2 – nμS, where Etot and EDSMoS2 are energies of P doped MoS2 and S-deficient MoS2, n and μS are the number of dopant P atoms and the chemical potential of S calculated by the binding energy of S2 molecule. The calculated results are –0.411, –0.906 and –2.151 eV for 1P-, 2P- and 3P-MoS2, respectively. The negative values indicate the doping of P is energetically favorable. Moreover, the formation energies decrease with the increasing of P dopant concentration, suggesting that P doping can be easily achieved in MoS2. We have added related discussions in the revised manuscript.

“First of all, we investigate the stability of P-doped MoS2 by calculating the formation energy as Eform = Etot – EDSMoS2 – nμS , where Etot and EDSMoS2 are energies of P doped MoS2 and S-deficient MoS2, n and μS are the number of dopant P atoms and the chemical potential of S calculated by the binding energy of S2 molecule. The calculated results are –0.411, –0.906 and –2.151 eV for 1P-, 2P- and 3P-MoS2, respectively. The negative values indicate the doping of P is energetically favorable. Moreover, the formation energies decrease with the increasing of P dopant concentration, suggesting that P doping can be easily achieved in MoS2.”

Reviewer 2 Report

This manuscript reports on the modulation of MoS2 properties by means of P doping in order to enhance its HER activity. The employed DFT-based theoretical approach reveals the hybridization between the Mo-4d and P-3p orbitals leading to large pz orbitals and, therefore, improves the activity of MoS2 in comparison with the pristine surface. Although the manuscript reports interesting results, I have the following concerns that I hope the authors can elaborate on them. 

First of all, selecting phosphorus as a dopant should be justified. 

In fig 1 (a), it is shown that the three equivalent sites of MoS2 hexagonal lattice are occupied by P atoms. This, I assume, would lead to a change of lattice parameters due to relaxation. Is it an energetically favorable model? 

My other concern is also related to substitutional doping as follows. Can the authors elaborate on the experimental realization of such heavy doping scenario? 

Why the band gap of pristine MoS2 is underestimated and yield \approx 1.7 eV instead of \approx1.9 eV? 

The potential of MoS2, together with other metal sulfide compounds in hydrogen evolution, has been reviewed in this very recent paper. I suggest the authors cite this paper. https://doi.org/10.3390/catal12111316

The manuscript has a serious flow problem, grammar and typo errors, and nonmentioned figures in the text, which must be solved in this stage of revision. I suggest the authors correct and answer the above-mentioned comments before my final decision. I might have further comments on the revised version as well.

Author Response

Response to Reviewer 2 Comments

This manuscript reports on the modulation of MoS2 properties by means of P doping in order to enhance its HER activity. The employed DFT-based theoretical approach reveals the hybridization between the Mo-4d and P-3p orbitals leading large pz orbitals, and therefore, improves the activity of MoS2 in comparison with the pristine surface. Although the manuscript reports interesting results, I have the following concerns that I hope the authors can elaborate on them. 

Thank the referee for the positive comments. 

Point 1. First of all, selecting phosphorus as a dopant should be justified.

Reply: Thank the referee for the comment. Phosphorus as a dopant doped into MoS2 has been proved to be feasible and reliable in both acid and alkaline conditions in experiments. For example, Li et al. have shown that P-doped MoS2 nanosheets with enlarged interlayer spacing exhibited remarkable electrocatalytic activity and good long-term operational stability [ACS Energy Lett., 2017, 2, 745]. Bian et al. have found P-doped MoS2 nanosheets on carbon cloths displayed excellent catalytic activity with a low overpotential of 133mV to drive the current density of 20 mA cm-2, along with good stability in acid media [ChemCatChem., 2018, 10, 1571]. Huang reported interface construction of P-substituted MoS2 was considered as an efficient and robust electrocatalyst for alkaline HER [Nano Energy, 2020, 78, 105253]. We have added related discussions in introduction section in the revised manuscript. 

“In fact, phosphorus as a dopant doped into MoS2 has been proved to be feasible and reliable in both acid and alkaline conditions in experiments. For example, Li et al. have shown that P-doped MoS2 nanosheets with enlarged interlayer spacing exhibited remarkable electrocatalytic activity and good long-term operational stability. Bian et al. have found P-doped MoS2 nanosheets on carbon cloths displayed excellent catalytic activity with a low overpotential of 133mV to drive the current density of 20 mA cm-2, along with good stability in acid media. Huang reported interface construction of P-substituted MoS2 was considered as an efficient and robust electrocatalyst for alkaline HER.”  

Point 2. In fig 1 (a), it is shown that the three equivalent sites of MoS2 hexagonal lattice are occupied by P atoms. This, I assume, would lead to a change of lattice parameters due to relaxation. Is it an energetically favorable model? 

Reply: Thank the referee for the comments. P doping induces structural changes. The figure below shows the 3P doped MoS2 of 4×4×1 supercell, the structure changes greatly, one of the P dopant pulls out from the surface. Compared to pure MoS2, the lattice parameters change from a = b = 12.65 Å to a = b = 12.62 Å, the bond length of Mo-S neighbouring P dopant changes from 2.43 Å to 2.38 Å. To investigate the stability of P-doped MoS2, we calculate their formation energies by Eform = Etot – EDSMoS2 – nμS, where Etot and EDSMoS2 are energies of P doped MoS2 and S-deficient MoS2, n and μS are the number of dopant P atoms and the chemical potential of S calculated by the binding energy of S2 molecule. The formation energy of 3P doped MoS2 is –2.151 eV. The negative value indicates the doping of P is energetically favorable and can be achieved in MoS2.

Point 3. My other concern is also related to substitutional doping as follows. Can the authors elaborate on the experimental realization of such heavy doping scenario.

Reply: Thank the referee for the comments. It has been reported that P doped into MoS2 could be realized in the experiments. Ye et al. reported a one-step approach to improve the HER activity of MoS2 and MoP via formation of a MoS2(1-x)Px (x = 0 to 1) solid solution [Adv. Mater. 2016, 28 (7), 1427]. Liu et al. successfully doped P atom into MoS2, the concentration of P dopant was up to 5.1% [ACS Energy Lett., 2017, 2, 745]. Here, our calculated P dopant concentrations are respectively 2%, 4% and 6%, which can almost be compared with the experiments. We have added the related discussions in the revised manuscript.   

“Considered the structural symmetry, one, two and three P atoms respectively doped in MoS2 by substituting intrinsic S atoms, which corresponds to a doping concentration of 2%, 4% and 6%, respectively, which may be achieved in the experiments. Ye et al reported a one-step approach to improve the HER activity of MoS2 and MoP via formation of a MoS2(1-x)Px (x =0 to 1) solid solution. Liu et al successfully doped P atom into MoS2, the concentration of P dopant was up to 5.1%.”

Point 4. Why the band gap of pristine MoS2 is underestimated and yield \approx 1.7 eV instead of \approx1.9 eV? 

Reply: Thank the referee for the comments. This is caused by the defect of GGA-PBE method itself, which underestimates the correlation between excited electrons, making the calculated values lower than the experimental value, but it does not affect the qualitative analysis of the results.

Point 5. The potential of MoS2, together with other metal sulfide compounds in hydrogen evolution, has been reviewed in this very recent paper. I suggest the authors cite this paper. https://doi.org/10.3390/catal12111316.

Reply: Thank the referee for the suggestion. We have cited this paper in the revised manuscript. 

Point 6. The manuscript has a serious flow problem, grammar and typo errors, and nonmentioned figures in the text, which must be solved in this stage of revision. I suggest the authors correct and answer the above-mentioned comments before my final decision. I might have further comments on the revised version as well.

Reply: Thank you for the reminding. We have checked manuscript and corrected the spelling and grammar carefully, and hope these modifications and improvements can make our manuscript more suitable for publication.

Round 2

Reviewer 1 Report

Thanks to the authors for considering and addressing all questions.

Author Response

Thank the reviewer for the recommendation of our work. 

Reviewer 2 Report

The authors should correct the reference list according to the journal's requirements. Also, references 5, 14, and 19 have been cited incorrectly. These must be corrected.

Author Response

Thank the reviewer for the reminding. We have corrected references 5, 14, and 19 and carefully revised the reference list according to the journal's requirements.  We hope the modifications can be suitable for publication.

Round 3

Reviewer 2 Report

the paper can be accepted for publication.